# Digital Image Representation by Atomic Functions: The Compression and Protection of Data for Edge Computing in IoT Systems

**DOI:** 10.3390/s22103751

**Published:** 2022-05-14

**Authors:** Viktor Makarichev, Vladimir Lukin, Oleg Illiashenko, Vyacheslav Kharchenko

**Affiliations:** 1Department of Computer Systems, Networks and Cybersecurity, National Aerospace University “KhAI”, 17, Chkalov Str., 61070 Kharkiv, Ukraine; v.makarichev@khai.edu (V.M.); v.kharchenko@csn.khai.edu (V.K.); 2Department of Information-Communication Technologies, National Aerospace University “KhAI”, 17, Chkalov Str., 61070 Kharkiv, Ukraine; v.lukin@khai.edu

**Keywords:** image representation, lossless image compression, image protection, atomic function, atomic wavelet, discrete atomic transform, security, privacy, IoT, edge computing

## Abstract

Digital images are used in various technological, financial, economic, and social processes. Huge datasets of high-resolution images require protected storage and low resource-intensive processing, especially when applying edge computing (EC) for designing Internet of Things (IoT) systems for industrial domains such as autonomous transport systems. For this reason, the problem of the development of image representation, which provides compression and protection features in combination with the ability to perform low complexity analysis, is relevant for EC-based systems. Security and privacy issues are important for image processing considering IoT and cloud architectures as well. To solve this problem, we propose to apply discrete atomic transform (DAT) that is based on a special class of atomic functions generalizing the well-known up-function of V.A. Rvachev. A lossless image compression algorithm based on DAT is developed, and its performance is studied for different structures of DAT. This algorithm, which combines low computational complexity, efficient lossless compression, and reliable protection features with convenient image representation, is the main contribution of the paper. It is shown that a sufficient reduction of memory expenses can be obtained. Additionally, a dependence of compression efficiency measured by compression ratio (CR) on the structure of DAT applied is investigated. It is established that the variation of DAT structure produces a minor variation of CR. A possibility to apply this feature to data protection and security assurance is grounded and discussed. In addition, a structure or file for storing the compressed and protected data is proposed, and its properties are considered. Multi-level structure for the application of atomic functions in image processing and protection for EC in IoT systems is suggested and analyzed.

## 1. Introduction

### 1.1. Motivation

Nowadays, digital data are an integral part of human life. Data are used in various technological, financial, economic, and social processes. Moreover, data are a base of particular sectors of industry; they also provide the emergence and explosive development of new ones.

Digital images have a special place among data. They are a key object of research and analysis of such directions as digital signal and image processing [1], computer vision [2], machine learning [3], etc. The results obtained are widely applied in security systems for smart homes [4] and environment monitoring systems [5], autonomous control [6], robotics [7], and so on. A separate important issue is the efficiency improvement of image generation, processing, and transmission in the context of implementing Internet of Things (IoT) systems in industry and human domains [8].

Technology development has already provided and continues to produce a lot of high-resolution digital images. For instance, using modern sensors, remote sensing digital images of a very high resolution can be acquired [9]. Smartphones are another example. They provide more than 1 trillion digital photos every year [10]. A lot of image databases have been collected. The main examples are the SICAS Medical Image Repository [11], the European Space Agency “Sentinel-2” database [12], and the computer vision dataset “CelebA” [13].

For this reason, the problem of reducing the resources required for image processing, storage, and transferring via networks is of particular interest. This aspect of image processing is complemented by several circumstances that are more important.

Firstly, for many applications, it is required to ensure data privacy, integrity, and confidentiality in parallel with their compression, transmission, and processing [14,15]. Data protection during image processing is carried out, as a rule, using standard additional software and hardware without considering the particularities of the compression and processing processes themselves. It would be great to ensure data protection by embedding security-related procedures into imaging processes [16,17].

Secondly, image processing and compression should be, as a rule, performed at the lower levels of the system hierarchy that can be associated with dew and fog computing [18,19]. The concept of edge computing (EC) has become one of the most constructive and attractive due to the ability to offload traffic and make more powerful and intelligent means (hardware and software) that are placed near data sources in IoT systems [18,19]. The importance of aspects of the application of edge computing in data processing engineering is confirmed in [20].

Nowadays, the market for autonomous vehicles consists primarily of unmanned military drones. Current research shows that by 2023 it will transform into one that is 60 percent civilian, including ground, sea, air, and space transportation and expand to 636 billion euros, more than 40 times its current size. According to the industrial and researchers’ calculations, autonomous vehicles will make up to 20 percent of the total vehicle market by 2030, and three out of four of them will be used as ground transportation [21,22].

Cybersecurity ensuring the autonomous transportation systems of all types is extremely important since its secureness influences the safety of people (e.g., in an urban area, suburbs, etc.) and the environment. In particular, cybersecurity should be provided as far as autonomous vehicles should transmit (store and handle) data between each other, thus being the part of constantly changing computing systems of the Internet of Things, considering the massive amount of data which should be transmitted over this flexible Internet of autonomous vehicles such as UAVs and specific vulnerabilities of protocols and IoT systems [23,24].

A promising way to cope with providing the cybersecurity of the autonomous transportation (mobility) systems of different types could be the application of ideas and methods of machine learning (ML) and artificial intelligence (AI) as well. In this case, the trustworthiness of the implemented algorithms of cybersecurity ensuring which are executed by autonomous vehicles of all types (ground, sea, air, and space transportation) should be of the highest level and depend mainly on the functional efficiency of ML and AI technologies applied to the on-board system of the particular autonomous vehicle and its communications (and probably stationary systems of traffic control/control rooms in case of urban area).

The potential cybersecurity risks connected to the AI and ML components are in charge of replicating tasks previously addressed by human drivers (e.g., making sense of the environment or making decisions on the behaviors of the vehicle) [23,24]. AI components do not obey the same rules as traditional software (ML techniques indeed rely on implicit rules that are grounded on the statistical analysis of large collections of data, thus enabling automation to reach unprecedented cognitive capabilities). Thus, it opens at the same time new opportunities for malicious actors, who can exploit the high complexity of AI systems to their advantage by using AI cognitive differences in decision-making compared to human beings.

Thus, there is a need to develop edge computing-based engineering for secure image processing and compression engineering by use of novel approaches and algorithms which could assure synergetic effect in practical implementation useful for both academia and industry sectors.

### 1.2. State of the Art and Objectives

A large number of image formats have been developed. Many of them, in particular, BMP, PNG, and JPEG, are widely used. Moreover, some of them are de-facto standards for special type images. For instance, JPEG is still the main standard for digital photos. Particular attention was paid to the solution of the problem of reducing memory costs in recent years [25]. A lot of image compression algorithms, which became the base of the corresponding graphic formats, have been developed and studied [26]. Nevertheless, not all of them are widely used and supported by software applications, in particular Internet browsers. An example is JPEG2000, which was developed to replace JPEG. Perhaps, one of the reasons is the fear of IT companies violating patent restrictions. Even though many patents have expired, it cannot be argued that currently, there is a trend toward the use of this algorithm. Potentially, an absence of some algorithm wide support by the community of developers and practitioners can be explained by the complexity of standards that results in the development of new technologies, which have simple program implementations and, hence, are easy to support (see, for example, the algorithm QOI [27]).

An important trend in recent decades is the explosive hardware development, creation, and wide use of smartphones and tablet PCs [28]. There is also a decrease in price and an increase in the performance of data storage devices, as well as the development of new storing technologies. Moreover, an improvement of telecommunication systems provides a wide usage of cloud storage services, for example, Google Drive, One Drive, Dropbox, etc. All of these have led to a shift in interest of the scientific community in other directions, especially toward artificial intelligence and machine learning. In addition, a huge increase in computing power allows the rapid development and practical implementation of projects that were previously considered to be related to the far future and even science fiction. The examples are different systems and services based on artificial networks [29].

The reverse side of the great development of technologies is an explosive increase in data volumes [30]. Furthermore, cyber threats and hacker activity have increased significantly, leading to dramatic damage [31]. Additionally, the risks of using personal data for criminal purposes have grown [32]. Therefore, the development of methods, which can solve the problem of protected data storage, is of particular importance. An important driver is also a necessity to satisfy some special regulatory restrictions concerning personal data (for instance, General Data Protection Regulation in the European Union [33]). One of the solutions to this problem is the application of encryption [34] and compression algorithms [25,26]. Data can be compressed and, after that, encrypted. However, this requires huge additional resources, especially when processing big data. Another way is to apply compression to data that is already encrypted. However, such an approach greatly restricts the range of applicable compression algorithms and affects their efficiency [26].

Further, since, in many cases, image recognition is resource-intensive, one of its development trends is in the reduction of complexity, especially when applying deep learning approaches [35,36]. Additionally, a satisfaction of green technologies principles and requirements [37] should be provided to reduce energy costs and, as a result, CO_2_ emission, as well as thermal pollution of the environment. One possible solution is to create and apply such a data representation, which would be ready for the direct application of recognition algorithms.

Therefore, the problem of developing an image representation, which provides compression and protection in combination with the possibility of direct application of recognition, retrieval, and classification algorithms, is of particular importance. The relevance of this problem is also confirmed by IT industry trends analysis, in particular, in the area of data storing [38,39] and protection [40,41]. Concerning digital images, trends such as complexity reduction, especially when applying deep learning techniques for image analysis and recognition, should be noted [19,20]. All of these take on special significance when applying edge computing (EC) approaches in the Internet of things (IoT) systems [18,42]. Indeed, modern mobile devices such as smartphones and tablet computers provide digital photos of a very high resolution. Further operating these data in EC-powered artificial intelligence systems, i.e., AIoT [20,43], as well as distributed machine learning (ML) systems in EC [44], requires huge traffic and computational resources. Additionally, it provides such issues as power waste, latency, etc. Furthermore, appropriate security and privacy protection must be ensured [45].

In the current paper, the possibility of solving this problem using atomic functions [46,47] is considered. Atomic functions were introduced in the early 1970s of the 20th century. The main reason for their development was a combination of necessity to solve numerically different applied problems with poor computational capabilities. This provided a search for constructive tools that were an alternative to algebraic and trigonometric polynomials. In this paper, atomic functions,
(1)$$up_{s}\left(x\right)=\frac{1}{2\pi }\int_{-\infty }^{\infty }e^{itx}\prod_{k=1}^{\infty }\frac{sin^{2}\left(st{\left(2s\right)}^{-k}\right)}{s^{2}t{\left(2s\right)}^{-k}sin\left(t{\left(2s\right)}^{-k}\right)}dt,$$
are used. A particularly important case of these functions is $up_{1}\left(x\right)$ which is a well-known up-function of V.A. Rvachev [46]. Atomic functions $up_{s}\left(x\right)$ have several useful features, which are important in the context of the problem stated above. Indeed, they have good approximation properties and provide fast data decomposition [46,47]. Their application to the development of image representation, which combines the features described, is promising. Moreover, the exact values of $up_{s}\left(x\right)$ can be computed using fast dynamic algorithms [48].

The main contribution of this work is a lossless image processing algorithm based on atomic functions $up_{s}\left(x\right)$ that has low time complexity and provides a combination of compression, data protection, as well as image representation, which can be used in recognition, classification, retrieval, and other artificial intelligence technologies directly without preprocessing steps. A novelty is an ability to vary a structure of discrete transform, which is based on $up_{s}\left(x\right)$ and constitutes a core of the algorithm considered, without significant impact on the efficiency measured by the compression ratio that is a classic performance indicator [26]. In Table 1, the related references are presented, and their features are discussed. In contrast to the works presented, a combination of image compression, protection, and convenient data representation is the focus of the current research.

This paper is organized as follows. First, the formulation of the problem and an approach to the current research are introduced. Second, a solution and algorithms of digital image compression based on atomic functions are presented. Next, efficiency research is carried out, and possible edge computing-based applications and possibilities for data protection are discussed. Finally, conclusions follow.

## 2. Formulation of the Problem and an Approach

In the current research, we consider raster digital images. This type of data can be considered a two-dimensional matrix [1]. Generally, the problem is to develop such a transformation of an image matrix that provides its further efficient compression, protection, and representation, which is convenient for recognition. Different ways to apply atomic functions (1) are discussed in [47]. It was shown that atomic wavelets, constructed using $up_{s}\left(x\right)$, are the most appropriate tool for discrete data processing, in particular, matrices. These functions are applied in discrete atomic compression (DAC) of digital images [49]. The algorithm DAC belongs to the class of lossy compression algorithms [26].

Figure 1 show the structure of DAC. Discrete atomic transform (DAT), which is the wavelet transform based on atomic functions $up_{s}\left(x\right)$, constitutes a core of the DAC algorithm. In [47], it was proved that DAT has linear time complexity, i.e., $T_{DAT}\left(N\right)=O\left(N\right),$ where *N* is the size of the data processed (in the case of matrix processing, *N* is a number of pixels). Moreover, it requires insignificant additional memory overhead. Furthermore, since spaces of atomic functions $up_{s}\left(x\right)$ are asymptotically extremal for the approximation of differentiable functions, i.e., these functions have good approximation properties, the DAT provides a representation of the image by a set of DAT coefficients, most of which are small-valued.

In other words, a digital image can be described well using just a small number of DAT coefficients that, in addition, can be found using low-complexity computations. For this reason, the DAT can be considered a procedure for dimension reduction, and image description by DAT coefficients is the representation that is convenient for further image analysis. This is similar to using the discrete cosine transform (DCT) [29]. An important advantage of DAT compared to DCT is computational complexity. Indeed, in general, the time complexity of DCT is $T_{DCT}\left(N\right)=O\left(N^{2}\right)$; although, in many practical cases, fast algorithms provide O(Nloglog N ) [50].

Next, by construction, DAT coefficients are real numbers and, in general, non-integers. At the same time, in most cases, elements of the mage matrix are integers. For instance, each element of matrices of 24-bit full-color digital images has three components: red, green, and blue, each of which is an integer from the range 0, 1, …, 255 [1]. To provide data compressing, quantization of DAT coefficients and further encoding using lossless compression methods are applied in DAC (see Figure 1) [49].

It is quantizing that makes this algorithm lossy, i.e., a decompressed image does not match the corresponding source. Nevertheless, the quality loss control mechanism, which ensures the desired loss of quality measured by maximum absolute deviation (MAD), root mean square error (RMSE), and/or peak signal-to-noise ratio (PSNR), was recently developed, and the modes of DAC, which can produce compressed images without visual distortions, were obtained [49]. Furthermore, it was shown that variation of the DAT structure provides a minor variation of DAC efficiency indicators. This feature of the DAC algorithm provides high-level protection for the digital images processed [49].

Hence, the DAT-based approach, which is used in DAC, is a way to develop such image representation that provides its compression and protection in combination with the convenience of further analysis, in particular, recognition, classification, retrieval, etc.

However, the following problem arises. When processing, for example, digital photos, some loss of quality without visual distortions can be considered acceptable. At the same time, even visually lossless compression can be unacceptable in some cases. For instance, when compressing medical data or images, which are supposed to be used as evidence, the changes may lead to incorrect results of analysis [2]. The current paper aims to answer the following question: is it possible to modify the algorithm DAC to provide a lossless compression of digital images? To answer this question, the following tasks are solved step by step:First, to develop an extension of lossy DAC that allows reconstruction of the image compressed without any distortions;Second, to investigate features of the constructed lossless image compression algorithm and determine parameters that affect its effectiveness;Third, to find such parameters of lossless DAC that provide its best performance in terms of image compression.

## 3. DAC Based Solutions

In this section, lossless discrete atomic compression of digital images is proposed, and its basic features are discussed.

### 3.1. Lossless DAC

The DAC algorithm (Figure 1) is constructed using the following approach, which is also applied in JPEG, JPEG2000, and other image compression algorithms [25,26]: preprocessing → discrete data transform → quantization → encoding. There are several ways to modify this classic scheme to obtain lossless compression. First, reversible procedures that provide exact source data reconstruction can be used at each processing step. However, it requires significant changes. For instance, in the algorithm JPEG2000, the lossless mode uses discrete wavelet and color space transforms that differ from the corresponding procedures of the lossy (conventional) one [25,26]. Such an approach can be implemented in DAC. In this case, the DAT should be modified or even replaced, which can result in a decrease in the compression efficiency of the method and its data protection features. Additionally, this makes the software implementation of DAC more complicated and, hence, more difficult to support.

Another way is to supplement the already existing version of DAC with additional steps that provide an absence of quality loss. The following idea seems to be one of the most simple and suitable methods to implement this. Let *A* be a matrix of a source image. Denote by *B* a matrix of the image that is reconstructed after compression by DAC. In general, these matrices are not identical. Consider a matrix *C = A − B*. It is suggested to compress this matrix using lossless methods and add the obtained data to the file that is provided by lossy DAC (Figure 2).

Thus, an extended DAC file, i.e., the file with a compressed image, consists of three main parts: a header, which contains, in particular, information about image size and parameters of lossy compression; a “lossy” part that contains compressed quantized DAT coefficients (actually, the data that provide the matrix *B*); the “extension” that provides the matrix *C*. It is clear that such an approach requires a modification of the DAC algorithm. Lossless image compression by DAC is shown in Figure 3. It can be seen that it consists of a lossy compression step, computation of difference matrix, and appending the “lossy” part with its encoded values.

At the first step of the presented algorithm, the RGB-to-YCrCb conversion is applied to the matrix of RGB components (*A*) of an image to be compressed. This step provides matrices *Y*, *Cr*, and *Cb* of luma and chroma components, respectively.

Then, the DAT procedure is applied to each of them, and three matrices of DAT coefficients are computed. After that, the values of the matrices obtained are quantized and encoded. This provides the “main data” of the extended DAC file (see Figure 2).

To obtain the “extension” part of the DAC file, inverse steps are used: elements of the matrices of quantized DAT coefficients are dequantized; then, inverse DAT and YCrCb-to-RGB conversion are applied, and the RGB matrix (*B*) of the reconstructed image is obtained. The difference matrix *C* = *A* − *B* is computed, and its values are encoded.

We stress that each of the steps applied has a time complexity that is linear concerning the number of the image pixels. Consider the encoding of values of matrix *C* in more detail.

To encode the matrix *C*, which defines a difference between the source image (*A*) and its “lossy” version (*B*), it is proposed to apply the same method that is used for lossless compression of quantized DAT coefficients, i.e., a combination of binary arithmetic coding with Golomb codes [49]. Elements of *C* are integers and are assigned to the bitstream using the following rule: zero elements are assigned to 0; each positive value $c_{ij}$ is assigned to the code
$$\underset{2k-1}{\underbrace{11\ldots 1}0}$$
where $k=c_{ij}$; and, finally, each negative value $c_{ij}$ is assigned to the code
$$\underset{2k}{\underbrace{11\ldots 1}0}$$
where $k=-c_{ij}$. After that, the bitstream obtained is compressed using binary arithmetic coding [26].

The proposed approach to lossless compression is based on several assumptions. First, it is assumed that the *CR* of lossy compression is quite large (let us say, at least larger than 2). Second, it is supposed that the *CR* for matrix *C* is quite large as well (sufficiently larger than unity) because the values of *C* are represented by a limited number of small integers. Then, one can expect that the aggregate *CR* can be essentially larger than unity. Furthermore, it is possible to assume that if the *CR* compressing for *B* increases, the *CR* for *C* decreases; thus, it should be some optimum.

In Figure 4, the decompression process is provided. It is obvious that the source image is reconstructed exactly. Indeed, the decompression procedure consists of three steps. The first and the second ones provide the matrices *B* and *C*, respectively. After that, these matrices are added, and *A = B + C*, which is a matrix of the original image, is computed. Using a part of the DAC file, which contains the “main data” (see Figure 2), matrix *B* is obtained. For this purpose, the decoding procedure is used, and matrices of quantized DAT coefficients are obtained. Then, inverse DAT and YCrCb-to-RGB conversion are applied, and matrix *B* is computed. After that, decoding of the rest of the DAC file, i.e., an “extension” part (see Figure 2), provides matrix *C*. By summing up this matrix with *B*, the digital image, which coincides with the source one, is obtained. Thus, the proposed image compression algorithm is lossless.

Further, features of the proposed modification of the algorithm DAC are considered, and its efficiency is investigated.

### 3.2. Features of Lossless DAC

In the proposed lossless DAC, the lossy DAC part, which consists of DAT, coefficient quantizing, and encoding, remains unchanged. Therefore, the possibility of varying the structure of DAT, which is important for data protection, also remains unchanged. Consider this possibility in more detail.

DAT is based on atomic wavelets that provide a decomposition of data given by an array into orthogonal components (see Figure 5). The number of orthogonal layers obtained is called a depth of transform [47]. Depending on how one-dimensional DAT is applied to blocks of a matrix, results of the different structures are obtained. In Figure 6, the matrix transform, which is denoted by DAT1, is shown. It is constructed as follows: first, the one-dimensional procedure DAT of the depth n is applied to each row of the given matrix; then, it is applied to each column of the matrix obtained. Further, in this paper, we consider the case *n* = 5.

Another approach denoted as DAT2 is given in Figure 7 and Figure 8. The DAT2 of depth 1 is defined as follows: first, the one-dimensional DAT of depth 1 is applied to each row of the matrix processed and then to each column of the matrix obtained (Figure 7). If this matrix transform is applied several times and each time just the upper left block, which contains low-frequency coefficients, is transformed, then the DAT2 of the depth n is obtained (Figure 8). Here, the depth n is several using the DAT2 of the depth 1. In this research, the cases *n* = 1, 2, 3, 4, 5 are considered. It is clear that the structure of the matrix of the DAT coefficients computed significantly depends on the structure of the applied matrix transform. Moreover, different mixes of DAT1 and DAT2 can be applied (Figure 9).

An appropriate inverse transform should be applied to reconstruct the source matrix. An absence of the direct transform specification makes reconstruction of the original data resource expensive, especially when processing a lot of high-resolution digital images. It is this feature that can be used to provide data protection. Additionally, DAT can be applied in combination with different partitioning schemes when the source matrix is divided into several parts and only after that the matrix transform is used (see, for example, algorithm ADCT [51]). Note that since the DAT of an array has computational complexity, which is linear in the size of the array processed [47], the complexity of the matrix transforms DAT1 and DAT2 is also linear:(2)$$T_{DAT1}\left(N\right)=O\left(N\right),$$
(3)$$T_{DAT2}\left(N\right)=O\left(N\right)$$
where *N* is several matrix elements, i.e., *N* is several pixels of an image processed. Note that the constants in the asymptotic expressions (2) and (3) are different, which may result in different expenses of time when processing the same matrix.

In [49], it was shown that the performance of the DAC algorithm depends on the structure of the matrix transform applied. From the results obtained, it follows that in most cases, a difference is minor. Note that it is lossy image compression using DAC that is studied in [49]. The following question arises: what is the dependence of the performance of the proposed lossless DAC on a structure of the DAT applied? Figure 10 show the results of one test photo compressing using DAC with DAT1 of the depth 5 and the algorithm ZIP, which is one of the most popular and widely supported data compression methods. The size of the BMP file with the test image is 11,528 KB. ZIP and DAC provide, respectively, 7744 KB and 5330 KB, which indicates that using DAC, it is possible to obtain better compression than using ZIP. Nevertheless, more complex research should be carried out. In Section 4, the corresponding study is presented.

Next, when applying DAC to lossy image processing, the results, in particular, the compression ratio and distortions, depend on quality loss settings, which are defined by coefficients of quantization [49,53]. In DAC, the parameter *UBMAD* defines the setting of quality loss:(4)$$UBMAD=\overset{}{\underset{i,j}{\sum }}\delta_{ij},$$
where the coefficients $\delta_{ij}$ are used in quantizing the elements of the blocks $B_{ij}$ of the matrix of DAT coefficients (see the Formulas (2), (3), and (8), (9) in [49]). Figure 10 show the compression result that is obtained for the case *UBMAD* = 95. It is proved in [49] that the variation of *UBMAD* leads to a variety of distortions. Hence, the difference between the source image and the reconstructed one depends on this parameter; therefore, the size of the file obtained using the proposed lossless DAC also depends on *UBMAD*. For this reason, the following question naturally arises: what is the value of *UBMAD* that provides the best compression? Section 4 answers this question.

Further, by construction, a unified approach to compressing both the quantized DAT coefficients and the elements of the difference matrix is used. This option provides a simple software implementation of the DAC algorithm. Indeed, its lossy mode should be supplemented with just one step that employs the encoding functions, which have already been developed, to obtain the lossless one. On the one hand, such unification can be useful in software implementation.

On the other hand, adaptive coding strategies can lead to higher compression efficiency. More complicated methods might provide better results. For instance, binary arithmetic coding, which is used in combination with complex data models and contexts, produces the great performance of the algorithms JPEG2000 [25,26], AGU [52], and ADCT [51]. Nevertheless, a complication of DAC may affect its further wide use and support.

Additionally, another useful feature is the possibility to reduce memory costs quickly due to switching from lossless compression to the lossy one. For this purpose, a part, which contains the compressed elements of the matrix of differences, i.e., the matrix *C*, can be deleted. Such an approach is of particular importance, especially if the “lossy” part of the DAC file (see Figure 2) contains data that provides a decompressed image of the acceptable quality. To illustrate this feature, consider the test photo provided in Figure 11a. It is a 544×393 full-color image. The size of the BMP file containing this photo is 626 KB. DAC with DAT1 of the depth 5 and *UBMAD* = 95 is applied.

The following results are obtained: the lossless and lossy modes provide compressed files of the size 312 KB and 59 KB, respectively. Note that lossless compression by ZIP (MS Windows 10 built-in function) produces 486 KB. In Figure 12b, the image, which is reconstructed after lossy compression by DAC, is presented. Comparing this image with the source one, it can be seen that there is no visual distortion. Additionally, quality loss metrics MAD and *PSNR* are computed: *MAD* = 18, *PSNR* = 40.9 dB, which indicate the high quality of the result obtained.

In Figure 12, representations of the considered test photo by different numbers of DAT coefficients are shown.

It is obvious that the image presented in Figure 12a has poor quality. It can be accepted only if an icon of the original image is required. Nevertheless, analyzing this image, the following conclusion can be made: Figure 12a quite possibly contains a face. More components provide more reliable recognition (see Figure 12b). Additionally, a small number of DAT coefficients corresponding to luma components of the image can be used to identify a person (Figure 12c). Finally, an increase in the number of DAT coefficients with a high probability provides the correct identification of the person (Figure 12d).

Thus, using the DAT procedure, it is possible to obtain a representation of digital images that satisfies the above requirements.

## 4. Efficiency of Lossless DAC

In the previous section, several particular examples of image processing using the proposed modification of the DAC algorithm were presented. It follows that its application to lossless image compression is promising. Nevertheless, to prove this statement, a more detailed analysis, which includes processing a greater number of test images, should be carried out. Further, an appropriate study is carried out.

### 4.1. Efficiency Analysis Approach

In the current research, a set of 24 test images from the database TID2013 is used [54]. These data were obtained from the Kodak lossless true color image suite [55]. Figure 13 show small copies of these pictures. Each of them is a 544×393 full-color image, saved in BMP format without compression. The size of any source file is 576 KB.

It is proposed to apply the following test data processing procedure:(1)Fix a structure of the procedure DAT; namely, DAT1 of the depth n=5 and DAT2 of the depth n=1, 2, 3, 4, 5 are considered;(2)Fix a value of the parameter UBMAD that defines coefficients of quantization;(3)Compress each test image by lossless DAC with the settings fixed and compute the compression ratio (*CR*):$$CR=\frac{size\;of\;source\;file}{size\;of\;compressed\;file}.$$

Further, it is suggested to determine the average, minimum, and maximum values of CR for each mode of DAC that is defined by the structure of DAT and the value of UBMAD.

The proposed approach provides, in particular, a dependence of *CR* on *UBMAD* for each structure of DAT considered. Additionally, the value of the parameter *UBMAD*, which results in the highest *CR*, can be determined.

### 4.2. Results of Test Data Processing

In Table 2, Table 3, Table 4, Table 5, Table 6 and Table 7, the results of the test data processing are given. Additionally, Figure 14 show the dependence of minimum, maximum, and average values of *CR* on the parameter *UBMAD* for each considered structure of DAT. In addition, in Table 8, the values of *UBMAD*, which provide the highest *CR*, are presented. Finally, a comparison of the total memory expenses required for storing the compressed and uncompressed data is presented in Table 9 and Figure 15 (each of the presented values is a sum of the sizes of image files of the corresponding format and is computed offline). Here, we note that the built-in functions are used in order to obtain ZIP compression, as well as images in PNG and TIFF formats.

## 5. Edge Computing-Based Application of DAC

### 5.1. Performance Analysis

From the results, which are presented in the previous section, it follows that the proposed lossless compression method provides sufficient memory cost reduction. We stress that this is achieved primarily due to the good approximation properties of the functions applied.

Further, for each considered structure of DAT, the dependence of the average (as well as minimum and maximum) value of CR on the parameter UBMAD is almost similar. It can be seen that CR as a function of UBMAD decreases for any considered $UBMAD>UBMAD_{max}$, where the values of $UBMAD_{max}$ are presented in Table 8. It can be explained as follows: a higher value of this parameter produces a greater loss of quality [25] and, hence, greater memory expenses are required for storing the differences (see the structure of DAC-file, Figure 2). For this reason, it is expected that the behavior of CR(UBMAD) remains unchanged on the set $\left[UBMAD_{max};+\infty \right)$.

Next, comparing the highest values of CR for each structure of DAT (see the data in Table 2, Table 3, Table 4, Table 5, Table 6 and Table 7), one can conclude that the difference is minor (actually, the difference does not exceed 5.2 percent). It is also confirmed by the data presented in Table 9. This implies that significant variation in the structure of DAT applied does not result in a great variation of lossless compression performance. It is expected that the application of various mixes of DAT1 and DAT2 (for example, Figure 9) also provide the same efficiency. As in the case of lossy compression by DAC, this feature can be used to provide privacy protection of the image processed since there are a huge number of structures of the procedure DAT.

Analysis indicates the existence of more than $10^{110}$ different image representations by DAT coefficients. Each of them is specified by the key that can be given by a sequence of bits. To obtain the correct content of the compressed image, an appropriate inverse DAT is required (see Figure 16). An absence of the key, which defines the correct decompression, makes unauthorized image viewing expensive in terms of the required computational resources. For this reason, we propose not to store a key, which is applied when compressing some images, in DAC files. Furthermore, the structure of the DAT applied cannot be uniquely determined by the size of this file. Meanwhile, we stress that the privacy protection features of the DAC algorithm require deeper investigation. This will be the object of further research.

If the requirement to ensure the lowest memory cost is the priority, especially when processing large datasets, then an appropriate structure of DAT must be used. In the case considered above, it is DAT2 of depth 3. Additionally, better compression can be obtained by improving the encoding procedure of DAC. For this purpose, the following modifications can be applied:(1)The implementation of context adaptive binary arithmetic coding (CABAC) [26]; here, we note that several data models and their construction may vary depending on the structure of DAT applied;(2)The usage of bit planes scanning of quantized DAT coefficient blocks; like in the algorithms JPEG2000 [25], ADCT [51], and AGU [52], this approach can be applied in combination with CABAC.

An efficient analysis of these propositions requires comprehensive research. Moreover, such modifications may increase the time and spatial complexity of DAC, which both have a negative impact, especially when implementing this algorithm in IoT devices.

We note that according to algorithmic taxonomy, which is presented for existing biometric privacy-enhancing techniques in [56], the proposed method of image protection belongs to representation-level approaches.

Furthermore, the proposed structure of the DAC file (Figure 2) is convenient for object recognition [2,3]. Indeed, a small amount of data, which defines the principal features of the image processed (see Figure 12), can be extracted from the compressed file in a fast manner. For this purpose, decompression of the whole file is not required. Additionally, if recognition or some other kinds of image analysis uses DAT coefficients to obtain the processing results, the application of inverse DAT is not required, which reduces complexity.

Moreover, computation of the image matrix is not “expensive” since the inverse DAT has linear time and space complexity [47]. So, the representation of an image by a set of DAT coefficients, which is used in combination with further compressing, is one of the solutions to the problem considered in this paper.

### 5.2. The Principal Steps of Image Processing by Atomic Functions

In Figure 17, the principal steps of image processing by atomic functions and their features are shown. Lossy DAC is presented. It can be easily modified to obtain a lossless mode of DAC. Note that, when operating digital images in an EC system, this process is performed on the device (smartphone, tablet PC, laptop, etc.), and can be considered a main preprocessing stage (see Figure 18). We stress that an application of the proposed DAT representation of the image provides the following:(1)Resource-efficient EC and training at the edge, which is of particular importance, for example, in intelligent transportation systems [57];(2)Easy implementation in fog EC and mobile EC architectures [18], as well as in “non-classic” ones, for instance, short supply circuit IoT [58];(3)Data protection, in particular, satisfies the so-called zero-trust principle, which belongs to the set of top trends [40]; a high level of protection and confidentiality is ensured by the great variety of settings, in particular, the atomic function applied in DAT and a structure of this core procedure, as well as several ways to encode quantized DAT coefficients; despite this, its comparison to other methods, for example, biometric security through visual encryption [59] and lightweight cryptographic algorithm [60], must be carried out;(4)Ability to construct artificial intelligence of things or AIoT systems [20], as well as that one which provides distributed learning, edge learning, and mobile intelligence [44].

Finally, from Table 9 and Figure 15, it follows that lossless DAC provides better compression than ZIP. This illustrates the fact that, in general, the algorithms which are developed to compress data of some special type, have higher performance than methods which can be applied to compressing data of any type [26]. Additionally, storing the test data in PNG and TIFF formats requires higher memory expenses than using DAC. Note that a very limited number of lossless image compression methods are applied. The results obtained show that DAC provides better compression than ZIP, PNG, and TIFF, which are widely supported. Nevertheless, a comparison of DAC to other lossless compression algorithms, which in most cases require specialized software tools, should be carried out. In particular, an ability to process and analyze digital images via directly manipulating the corresponding compressed data is of interest, especially when applying systems with low computational capabilities. This will be a topic of another research.

### 5.3. Application of Atomic Functions in Image Processing for EC in IoT Systems

The IoT context raises the issue of information compression and protection even more, as it requires reducing the load on communication channels and ensuring cybersecurity in data transmission. The introduction of EC just reduces the severity of these problems by focusing preprocessing on nodes. In terms of security, this shortens the chain of vulnerable elements and stages of transformation, although it requires more careful protection of EC nodes. Additionally, the limited capabilities of elements of the edge devices layer (see Figure 18) affect the applicability of data processing, including image compressing. Indeed, on the one hand, compression makes image files smaller and, hence, fewer resources are required for storing and transferring data between IoT system layers.

On the other hand, any high-resolution image processing is resource-intensive and reduces the time between charging or replacing the device battery. For this reason, the usage of compression algorithms, including DAC, must take into account the specifics of the IoT system application. For instance, if an edge device is used as a sensor, and all other data manipulations are performed by elements of other layers, then extremely low complexity algorithms (for example, QOI [27]) should be applied. The algorithm DAC should be preferred if a combination of compression and protection features is required. Moreover, the image can be well-presented by a small number of DAT coefficients (for example, see Figure 12) that allow the implementation of AI/ML technologies in edge devices. Furthermore, if such data processing is deployed at other layers, their load and communication networks can be reduced since just a limited number of DAT coefficients must be transferred and then processed. Furthermore, as mentioned above, matrices of DAT coefficients have a block structure (Figure 6, Figure 7 and Figure 8). Each of these blocks can be processed and analyzed by elements of edge and cloud layers independently, which speeds up obtaining results and their further usage.

In addition, the following question naturally arises: can the approach considered be applied to provide lossless image compression by other discrete transforms, in particular, DCT, and can a better performance be obtained in this case? Any lossy compression algorithm can be appended by encoding the difference between the source image and the decompressed one. In other words, the method applied in DAC can be used in a wide range of data compression techniques. The efficiency of the algorithm obtained in this way significantly depends on the functional properties of the constructive tools applied. From this point of view, DCT is one of the most suitable transforms due to the energy compaction property and extremeness of trigonometric polynomials {cos(kx), sin(kx)} for the approximation of different functional classes [50].

Moreover, it provides the data representation that is widely used in different artificial intelligence and computer vision applications [29]. Nevertheless, the computational complexity of this procedure, as it is mentioned in Section 2, may impose some restrictions, especially when processing time requirements are very strict.

## 6. Conclusions

### 6.1. Discussion

It was shown in this paper that lossless image compression by the DAC algorithm with different structures of DAT, which is its core, can be obtained. Additionally, the current research has proved that variation of the DAT structure provides insignificant variation in the performance of the lossless compression method obtained.

As it was discussed, image representation by DAT coefficients, which is a description by atomic functions, can be considered a solution to the problem of such image representation development that provides compression and protection features in combination with ease of further analysis and object recognition. Meanwhile, other ways, which provide better efficiency in terms of different indicators, may exist.

The results of test data processing have shown that by using lossless DAC, it is possible to outperform the algorithm ZIP, which is one of the most popular data compression methods. However, a comparison with other lossless algorithms has not been made. It is supposed to be carried out in further research. Additionally, it is planned to modify the encoding procedure to improve the compression efficiency.

In addition, the great variety of structures of DAT provide a wide range of ways to represent data, in particular, digital images, and, hence, to develop different artificial intelligence methods while maintaining the uniformity of the construction approach. This feature may be useful, especially when creating systems which are resistant to attacks. It can be ensured by the generation and controlled choice of different DAC parameters, such as the structure of the procedure DAT applied and the atomic function $up_{s}\left(x\right)$ that defines this discrete transform.

### 6.2. Future Research

Hence, an application of the proposed image representation by atomic functions to IoT systems, which are used in combination with EC and AI, is promising. Future research can be related to the spatial complexity reduction of the DAC algorithm and the development of methods and algorithms for the multi-parametrical presentation and implementation of atomic functions to decrease the risks of successful attacks on the confidentiality and integrity of IoT systems, such as autonomous systems based on UAV fleets [61] and ground robots [62] for pre- and post-monitoring severe accidents of NPPs and other objects of critical energy, transport, and industrial infrastructures.

Within the framework of the study, no specific cases were considered for the most common methods used in UAV or UAV fleets/swarms, but only for this task. However, in our opinion, they can be fully adapted to improve the nomenclature of the airborne apparatus of Internet of Drones technology and Flying Edge Platforms [63]. Such an adaptation is useful in additional assessments of the complexity of the algorithms of compression and refinement for the image of a specific subject area.

## Figures and Tables

**Figure 1 sensors-22-03751-f001:**
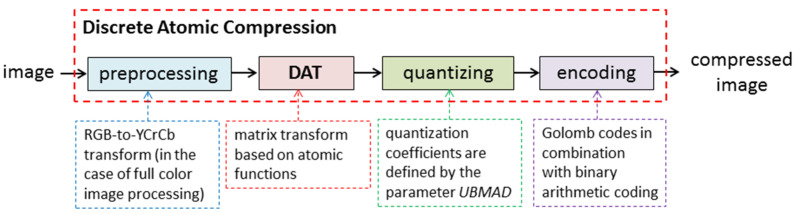
Discrete atomic compression of digital images.

**Figure 2 sensors-22-03751-f002:**
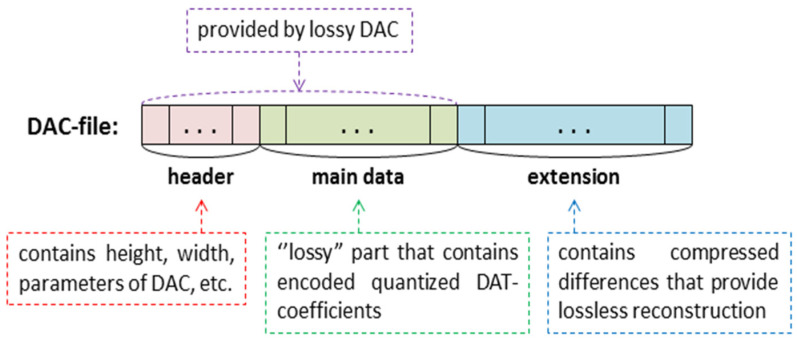
Extension of DAC file.

**Figure 3 sensors-22-03751-f003:**
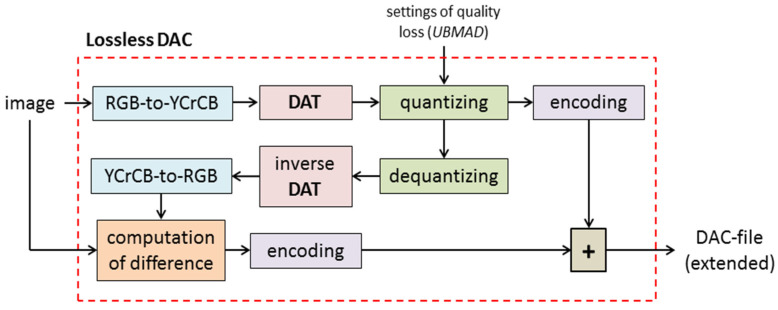
Lossless DAC: compression of the full-color digital image.

**Figure 4 sensors-22-03751-f004:**
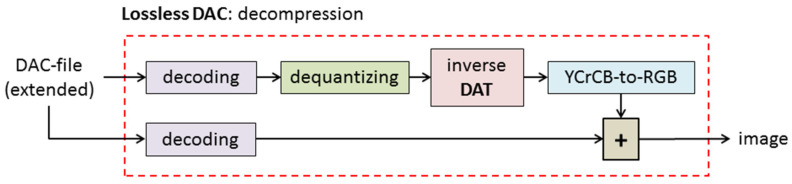
Lossless DAC: decompression of full-color digital image.

**Figure 5 sensors-22-03751-f005:**
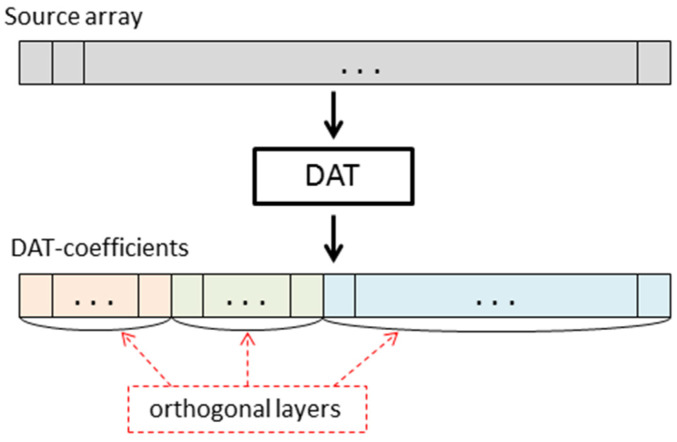
Discrete atomic transform of an array.

**Figure 6 sensors-22-03751-f006:**
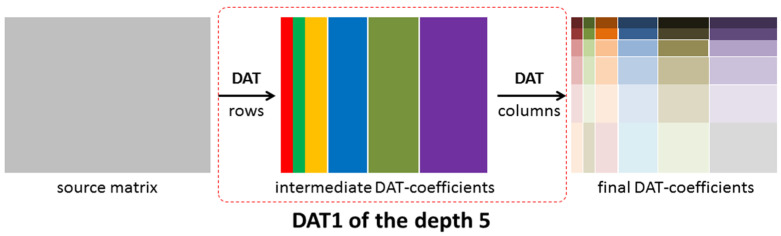
A structure of the matrix transform DAT1 of the depth 5.

**Figure 7 sensors-22-03751-f007:**
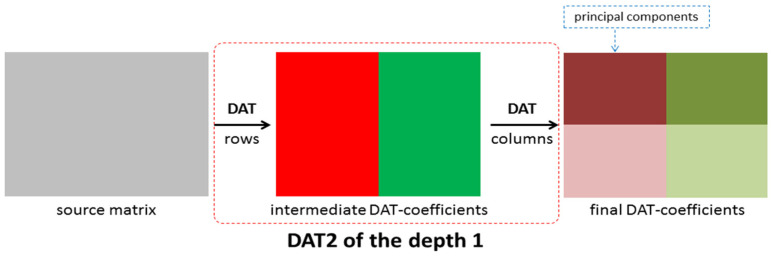
A structure of the matrix transform DAT2 of the depth 1.

**Figure 8 sensors-22-03751-f008:**
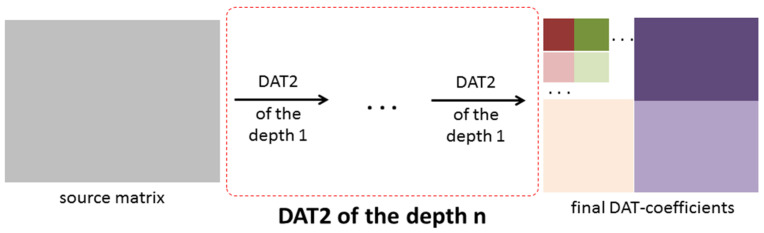
A structure of the matrix transform DAT2 of the depth n.

**Figure 9 sensors-22-03751-f009:**
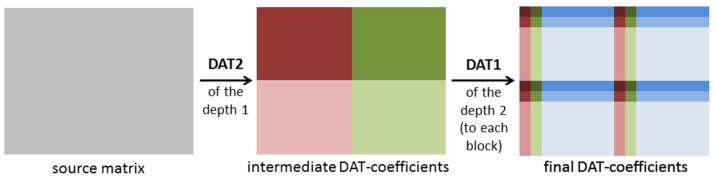
Mix of DAT1 and DAT2.

**Figure 10 sensors-22-03751-f010:**
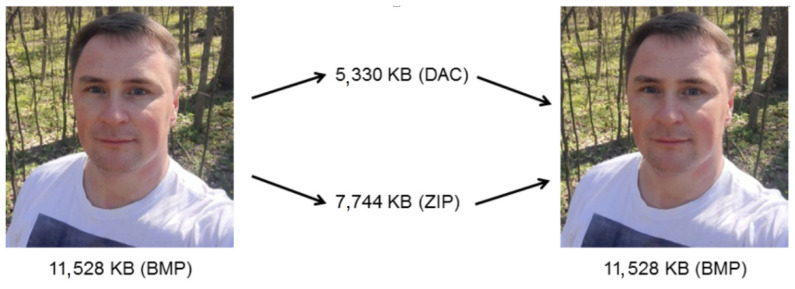
Compression of the test photo by ZIP and lossless DAC with DAT1 of the depth 5.

**Figure 11 sensors-22-03751-f011:**
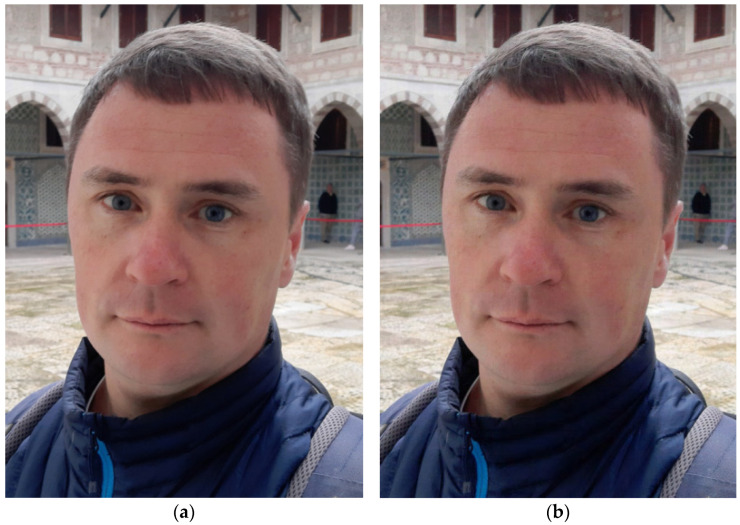
A 24-bit full-color digital image, 544 × 393, 626 KB (BMP): original (**a**); reconstructed after lossy compression using DAC with DAT1 of the depth 5 (UBMAD = 95) (**b**).

**Figure 12 sensors-22-03751-f012:**
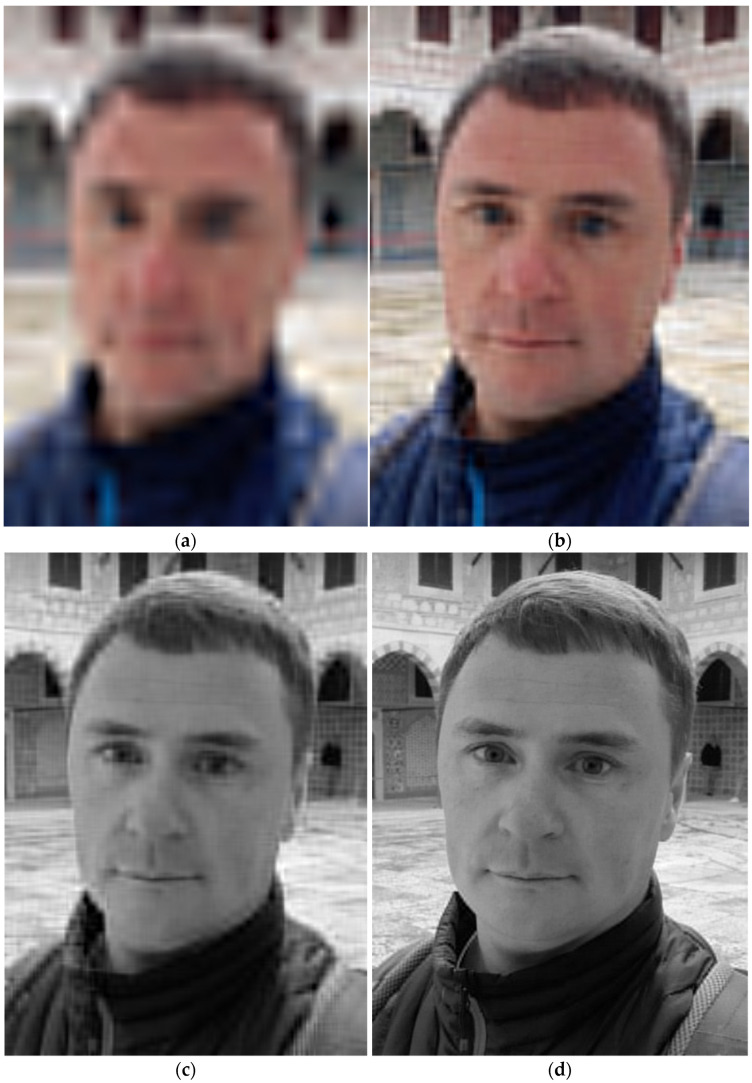
Representation of the image shown in Figure 11a by different number of DAT-coefficients: 0.34 percent of all values (**a**); 1.32 percent of all values (**b**); 1.73 percent of all components (**c**); 11.2 percent of all components (**d**).

**Figure 13 sensors-22-03751-f013:**
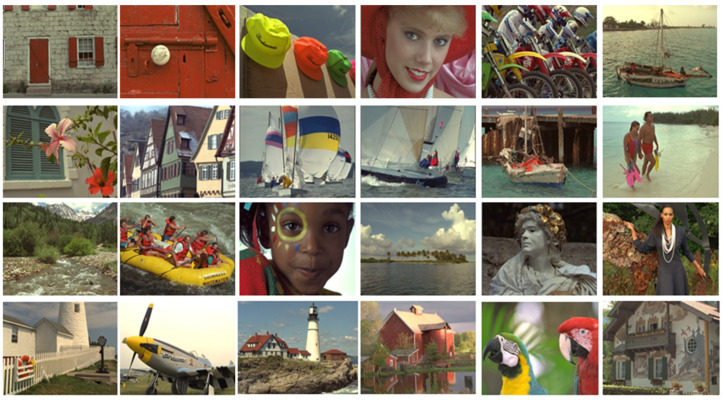
Small copies of test images.

**Figure 14 sensors-22-03751-f014:**
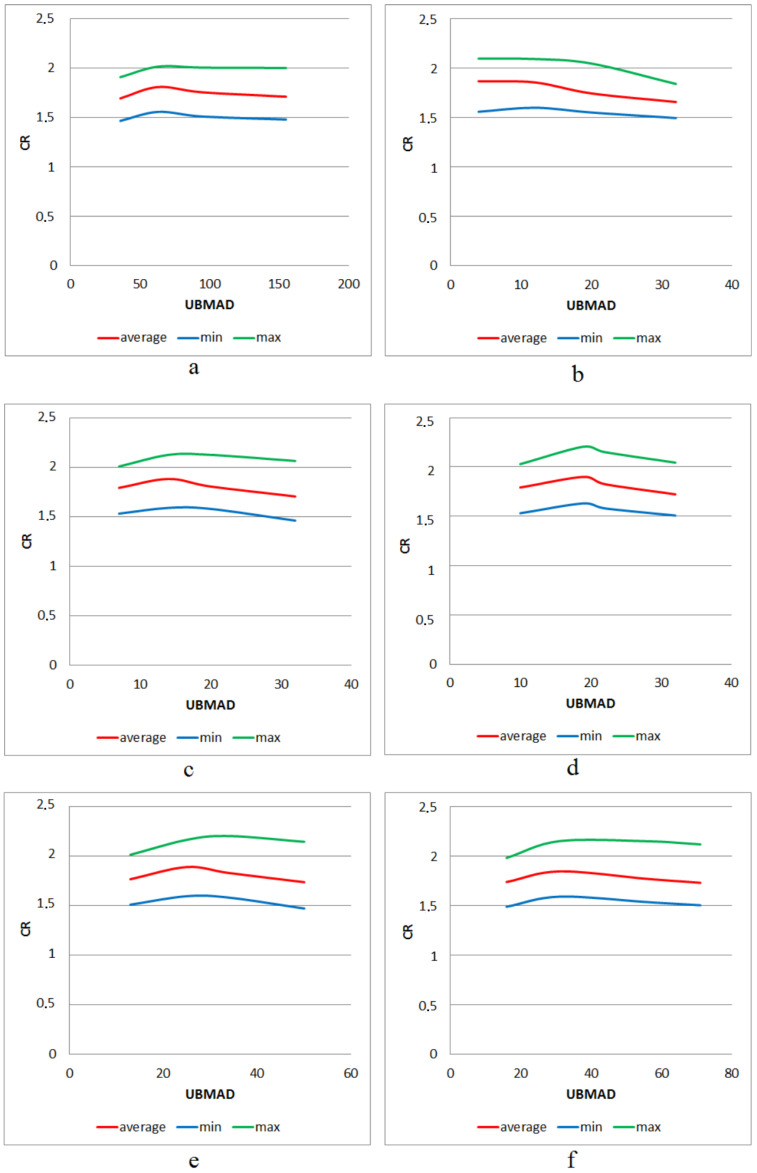
Dependence of the minimum, maximum, and average values of *CR* on *UBMAD* for different structures of the procedure DAT: DAT1 of depth 5 (**a**); DAT2 of depth 1 (**b**); DAT2 of depth 2 (**c**); DAT2 of depth 3 (**d**); DAT2 of depth 4 (**e**); DAT2 of depth 5 (**f**).

**Figure 15 sensors-22-03751-f015:**
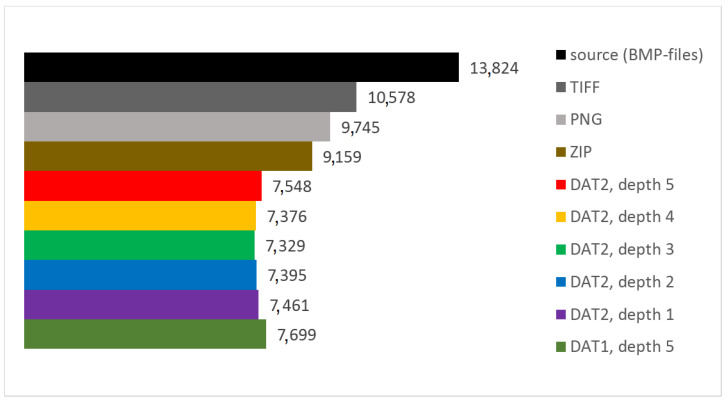
Total memory expenses required for storing the compressed and uncompressed data.

**Figure 16 sensors-22-03751-f016:**
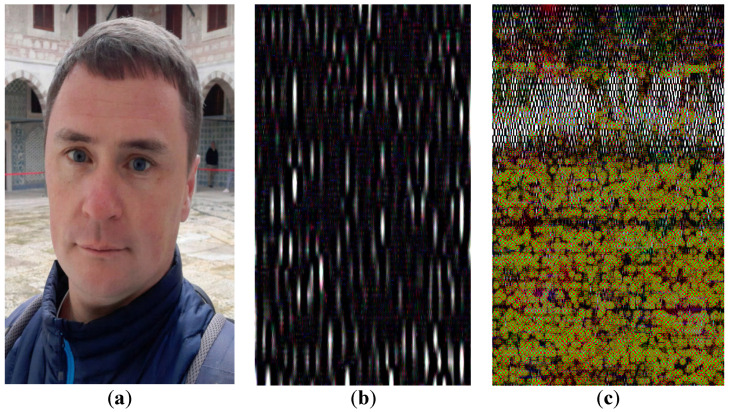
Decompression of the image given in Figure 11a: correct (**a**); incorrect (**b**,**c**).

**Figure 17 sensors-22-03751-f017:**
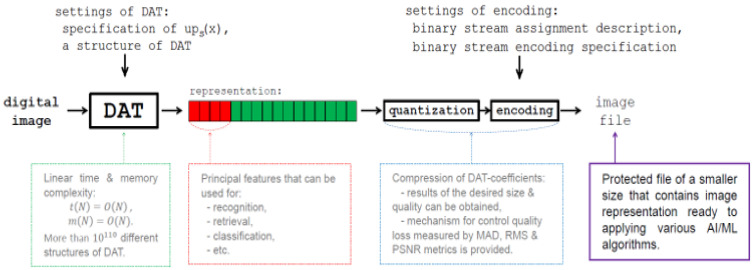
The principal steps of image processing by atomic functions.

**Figure 18 sensors-22-03751-f018:**
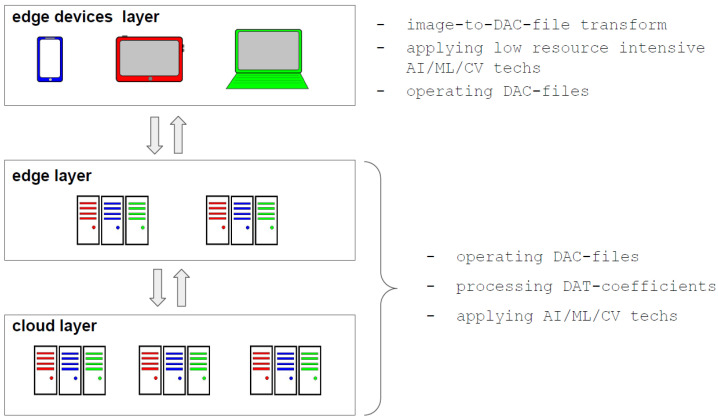
Application of atomic functions in image processing for EC in IoT systems.

**Table 1 sensors-22-03751-t001:** Summary of the related works.

Source	Features
Y.-Q. Shi and H. Sun [25]	(1) fundamentals of data compression are given; (2) image and video compression is a focus; (3) a brief description of data protection is provided.
K. Sayood [26]	(1) data compression fundamentals and principles are presented; (2) basic data algorithms are considered; (3) insufficient attention is paid to data protection functions.
QOI [27]	(1) a fast lossless image compression algorithm and the corresponding image format are given; (2) data protection property is not provided; (3) an ability to process and analyze the image compressed without decompression requires investigation.
V. Makarichev, V. Lukin and I. Brysina [47]	(1) $\mathrm{ways}\;\mathrm{to}\;\mathrm{apply}\;\mathrm{atomic}\;\mathrm{functions}\;up_{s}\left(x\right)$ to data processing are discussed, and the complexity of the corresponding algorithms is studied; (2) data compression and protection features are not a focus of this paper.
V. Makarichev and V. Kharchenko [48]	(1) $a\;\mathrm{dynamic}\;\mathrm{algorithm}\;\mathrm{for}\;\mathrm{computation}\;\mathrm{of}\;\mathrm{values}\;\mathrm{of}\;up_{s}\left(x\right)$ is developed, and its complexity is studied; (2) the results obtained provide various applications of these functions, in particular, data processing; (3) data compression and protection features are not a focus of this paper.
V. Makarichev, I. Vasilyeva, V. Lukin, B. Vozel, A. Shelestov and N. Kussul [49]	(1) $\mathrm{performance}\;\mathrm{of}\;\mathrm{lossy}\;\mathrm{image}\;\mathrm{compression}\;\mathrm{based}\;\mathrm{on}\;\mathrm{atomic}\;\mathrm{functions}\;up_{s}\left(x\right)$ is studied; (2) data protection feature is discussed; (3) lossless image compression is not considered.
C.K. Chui and Q. Jiang [50]	(1) fundamental constructive tools, which are applied in data compression, are presented, and their applications are given; (2) image representation by trigonometric polynomials and wavelets is discussed; (3) data protection feature is not a focus.
ADCT [51]	(1) the algorithm ADCT, which is based on DCT, is presented, and its performance is studied; (2) lossless compression and data protection features are not discussed.
AGU [52]	(1) the algorithm ADCT, which is based on DCT, is presented, and its performance is studied; (2) lossless compression and data protection features are not discussed.

**Table 2 sensors-22-03751-t002:** Dependence of the minimum, maximum, and average values of *CR* on *UBMAD* for the case of DAT1 of depth 5.

*UBMAD*	Min (*CR*)	Average (*CR*)	Max (*CR*)
36	1.4651	1.6922	1.9062
63	1.5557	1.8066	2.0128
95	1.5091	1.7537	2.0029
155	1.4781	1.7093	1.9989

**Table 3 sensors-22-03751-t003:** Dependence of the minimum, maximum, and average values of *CR* on *UBMAD* for the case of DAT2 of depth 1.

*UBMAD*	Min (*CR*)	Average (*CR*)	Max (*CR*)
4	1.5572	1.8656	2.0937
12	1.5979	1.8535	2.0896
20	1.5498	1.7427	2.0441
32	1.4934	1.6557	1.8381

**Table 4 sensors-22-03751-t004:** Dependence of the minimum, maximum, and average values of *CR* on *UBMAD* for the case of DAT2 of depth 2.

*UBMAD*	Min (*CR*)	Average (*CR*)	Max (*CR*)
7	1.5313	1.7915	2.0071
14	1.5895	1.8818	2.1247
20	1.5789	1.8053	2.1248
32	1.4607	1.7048	2.0633

**Table 5 sensors-22-03751-t005:** Dependence of the minimum, maximum, and average values of *CR* on *UBMAD* for the case of DAT2 of depth 3.

*UBMAD*	Min (*CR*)	Average (*CR*)	Max (*CR*)
10	1.5307	1.7920	2.2029
19	1.6311	1.8996	2.2062
22	1.5803	1.8256	2.1508
32	1.5073	1.7224	2.0439

**Table 6 sensors-22-03751-t006:** Dependence of the minimum, maximum, and average values of *CR* on *UBMAD* for the case of DAT2 of depth 4.

*UBMAD*	Min (*CR*)	Average (*CR*)	Max (*CR*)
13	1.5091	1.7653	2.0112
25	1.5944	1.8879	2.1621
34	1.5799	1.8282	2.2003
50	1.4704	1.7355	2.1422

**Table 7 sensors-22-03751-t007:** Dependence of the minimum, maximum, and average values of *CR* on *UBMAD* for the case of DAT2 of depth 5.

*UBMAD*	Min (*CR*)	Average (*CR*)	Max (*CR*)
16	1.4880	1.7371	1.9795
31	1.5899	1.8439	2.1490
56	1.5349	1.7685	2.1474
71	1.5029	1.7289	2.1159

**Table 8 sensors-22-03751-t008:** Values of the parameter $UBMAD_{max}$ that provide the highest *CR* for each structure of DAT.

Structure of DAT	*UBMAD* _max_
DAT1 of depth 5	63
DAT2 of depth 1	4
DAT2 of depth 2	14
DAT2 of depth 3	19
DAT2 of depth 4	25
DAT2 of depth 5	31

**Table 9 sensors-22-03751-t009:** Total memory expenses required for storing the compressed and uncompressed data.

Compressor	Memory Expenses, KB
DAC with DAT1 of depth 5	7699
DAC with DAT2 of depth 1	7461
DAC with DAT2 of depth 2	7395
DAC with DAT2 of depth 3	7329
DAC with DAT2 of depth 4	7376
DAC with DAT2 of depth 5	7548
ZIP of source	9159
PNG	9745
TIFF	10,578
source (BMP-files)	13,824

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
