# Peer review of "Digital Image Representation by Atomic Functions: The Compression and Protection of Data for Edge Computing in IoT Systems"

_sensors, 2022, doi:10.3390/s22103751_

Round 1

Reviewer 1 Report

The authors proposed to apply discrete atomic transform (DAT) that is based on a special class of atomic functions generalizing the well-known up-function of V.A. Rvachev. A lossless image compression algorithm based on DAT is developed and its performance is studied for different structures of DAT. It is shown that a sufficient reduction of memory expenses can be obtained. Also, a dependence of compression efficiency measured by compression ratio (CR) on the structure of DAT applied is investigated. It is established that variation of DAT structure produces a minor variation of CR. A possibility to apply this feature to data protection and security assurance is grounded and discussed. In addition, a structure of file for storing the compressed and protected data is proposed and its properties are considered. Multi-level structure for application of atomic functions in image processing and protection for EC in IoT systems is suggested and analyzed.
The proposed work is interesting and can be accepted for publication. I have the following comments.

1- Please underscore the scientific value added of your paper in your abstract.
2- In Figure 2, and Figure 3, it will be better and more clear if the authors are expressing about the steps in these figures as algorithms and explain them in more detail. 
3-  Please explain how did you calculate the total memory expenses? Is it calculated during the execution the proposed system or in offline.
4- The authors require to discuss their results in more details.
5- A tradeoff between power consumption at the edge gateway in compression and data saved over the IoT network is required. Can you discuss this point, please?. 
 6- The authors need to discuss the effect of reducing data on the latency.

Reviewer 2 Report

The work deals with a very interesting topic that is the edge computing. The work in this paper shows that the lossless image compression by the DAC algorithm with different structures of DAT, which is its core, can be achieved. Also, the present work has demonstrated that variation of the DAT structure offers an insignificant variation of performance of the lossless compression method attained.

The outcomes of examination data processing have exposed that with lossless DAC, it is possible to outdo the algorithm ZIP. But, a comparison with other lossless algorithms has not been done. The paper is very well written, clear, it contains a lot of information on the field of edge computing, I think it will be a good support for researchers in the field.

  1. The authors are invited to discuss the encoding procedure to improve the compression efficiency.
  2. The paper should make a comparison with other lossless compression algorithms
  3. Another thing, did the authors test the proposed algorithm on images in the case of using UAVs?

Reviewer 3 Report

I have several concerns regarding this paper. 

1. Please revise the grammatical issues of the paper. 
2. Add a summary table for Section 1.2 containing the limitation of existing works. Please highlight novelty in comparison with existing works. 
3. The paper mentioned IoT and edge computing. However, it is hardly used in the paper. The author could remove them from the title or give them more roles in the system design.
4. Please give reference to Eq. 1.
5. Comparison with existing works are missing. 

Round 2

Reviewer 3 Report

I am recommending to accept this paper.